# Decreased susceptibility of *Plasmodium falciparum* to both dihydroartemisinin and lumefantrine in northern Uganda

Patrick K. Tumwebaze[1,5], Melissa D. Conrad ●[2,5] ✉, Martin Okitwi[1], Stephen Orena[1], Oswald Byaruhanga[1], Thomas Katairo[1], Jennifer Legac[2], Shreeya Garg[2], David Giesbrecht ●[3], Sawyer R. Smith[3], Frida G. Ceja[4], Samuel L. Nsobya[1], Jeffrey A. Bailey ●[3], Roland A. Cooper[4] & Philip J. Rosenthal ●[2] ✉

Artemisinin partial resistance may facilitate selection of *Plasmodium falciparum* resistant to combination therapy partner drugs. We evaluated 99 *P. falciparum* isolates collected in 2021 from northern Uganda, where resistance-associated PfK13 C469Y and A675V mutations have emerged, and eastern Uganda, where these mutations are uncommon. With the ex vivo ring survival assay, isolates with the 469Y mutation (median survival 7.3% for mutant, 2.5% mixed, and 1.4% wild type) and/or mutations in Pfcoronin or falcipain-2a, had significantly greater survival; all isolates with survival >5% had mutations in at least one of these proteins. With ex vivo growth inhibition assays, susceptibility to lumefantrine (median $IC_{50}$ 14.6 vs. 6.9 nM, $p < 0.0001$) and dihydroartemisinin (2.3 vs. 1.5 nM, $p = 0.003$) was decreased in northern vs. eastern Uganda; 14/49 northern vs. 0/38 eastern isolates had lumefantrine $IC_{50} > 20$ nM ($p = 0.0002$). Targeted sequencing of 819 isolates from 2015–21 identified multiple polymorphisms associated with altered drug susceptibility, notably PfK13 469Y with decreased susceptibility to lumefantrine ($p = 6 \times 10^{-8}$) and PfCRT mutations with chloroquine resistance ($p = 1 \times 10^{-20}$). Our results raise concern regarding activity of artemether-lumefantrine, the first-line antimalarial in Uganda.

Drug resistance has challenged the treatment and control of falciparum malaria for decades[1]. Partial resistance to artemisinins, the most important drugs now used to treat malaria, was identified in southeast Asia about 15 years ago[2,3]. The partial resistance phenotype entails delayed clearance of parasites after treatment of patients with artemisinins[4], or enhanced survival of cultured parasites after exposure to dihydroartemisinin (DHA)[5]. Partial resistance in southeast Asia has been shown to be associated with any of about 20 candidate or validated mutations in the propeller domain of the *P. falciparum* kelch

13 (PfK13) protein[6,7]. Additional polymorphisms have accompanied PfK13 mutations in artemisinin resistant southeast Asian parasites[8,9] and in vitro selection of African parasites with DHA led to delayed clearance associated with mutations in *P. falciparum* coronin (Pfcoronin)[10]. PfK13 mutations validated as resistance mediators in southeast Asia have been reported in some other parts of the world[11–13], but until recently evidence of sustained prevalence of these mutations or documented clinical or in vitro resistance has been lacking[14]. Of greatest concern has been potential emergence or spread of resistance

[1]Infectious Diseases Research Collaboration, Kampala, Uganda. [2]University of California, San Francisco, CA, USA. [3]Brown University, Providence, RI, USA. [4]Dominican University of California, San Rafael, CA, USA. [5]These authors contributed equally: Patrick K. Tumwebaze, Melissa D. Conrad. ✉e-mail: melissa.conrad@ucsf.edu; philip.rosenthal@ucsf.edu

to Africa, where the bulk of malaria morbidity and mortality is seen[15]. Recently, PfK13 mutations previously linked to artemisinin partial resistance in Asia have been described in East Africa, with emergence of the PfK13 R561H mutation in Rwanda[16–18] and the C469Y and A675V mutations in northern Uganda[19–22]. In limited studies the C469Y, R561H, and A675V mutations have been shown to associate with artemisinin partial resistance measured clinically (delayed clearance) or in vitro (enhanced survival)[17,22], but our understanding of the impacts of the recently emerged *P. falciparum* mutations on drug sensitivity and the treatment efficacies of leading artemisinin-based combination therapies (ACTs) in Africa is incomplete.

ACTs combine a rapid acting artemisinin plus a slower acting partner drug[23]. In southeast Asia, the emergence of artemisinin partial resistance was followed by evidence for decreased antimalarial activity of the ACT partner drugs mefloquine[24] and piperaquine[25]. Notably, resistance to artemisinins, associated primarily with the PfK13 580Y mutation, and piperaquine, associated with novel mutations in PfCRT and *plasmepsin 2/3* gene amplification, led to failure rates for DHA-piperaquine >50% in Cambodia[26–28]. The recent emergence of partial resistance to artemisinins in East Africa may similarly lead to selection of parasites with decreased sensitivity to ACT partner drugs. Considering the massive toll of malaria in Africa, this consequence could be devastating, as was seen with the emergence of chloroquine resistance in the 1980s, leading to millions of excess malaria deaths[29].

The most commonly used ACT to treat malaria in Africa is artemether-lumefantrine. Unlike most antimalarials, lumefantrine is not available as a monotherapy, but rather is supplied only in combination with artemether. Perhaps for this reason, *P. falciparum* resistance to lumefantrine has not been definitively identified in parasites isolated in the field. However, an unstable lumefantrine resistance phenotype was selected by long-term incubation with the drug in vitro[30], and variations in susceptibility to lumefantrine have been seen in studies of field isolates. Lumefantrine susceptibility has been associated with polymorphisms in PfMDR1, a putative drug transporter. The PfMDR1 86Y mutation was selected by use of amodiaquine, selected against by therapy with artemether-lumefantrine, and has decreased to very low prevalence in most of Africa[15], including Uganda[20], in recent years. The wild type PfMDR1 N86 genotype is associated with lower ex vivo lumefantrine susceptibility compared to that in 86Y mutant parasites[31,32], and was associated with treatment failure in an analysis of data from 31 clinical trials[33], but changes in susceptibility associated with this allele were modest. Lumefantrine sensitivity is also decreased with *pfmdr1* gene amplification[34,35], but this polymorphism has been very uncommon in studied African parasites[15]. In Uganda, lumefantrine susceptibility decreased modestly in conjunction with loss of the PfMDR1 86Y mutation over time[31,32,36,37], the relative clinical efficacy of artemether-lumefantrine compared to that of artesunate-amodiaquine decreased over time[38], and a recent trial showed corrected treatment efficacy of artemether-lumefantrine below 90% at one site[39]. Overall, although in vitro activity of lumefantrine and clinical efficacy of artemether-lumefantrine have generally remained good, there is evidence that activity may be decreasing.

The recent emergence of artemisinin resistant *P. falciparum* in northern Uganda raises concern that resistance to ACT partner drugs may follow, as seen in southeast Asia. To explore this possibility and better characterize artemisinin partial resistance in Uganda, we compared genotypes and phenotypes of parasites collected in northern Uganda, where the PfK13 469Y and 675V mutations have emerged in recent years, and eastern Uganda, where we have studied clinical isolates for over a decade, and markers of artemisinin resistance have been uncommon. In addition, we analyzed a larger bank of phenotyped isolates from eastern Uganda to better explore genetic correlates of drug susceptibility phenotypes.

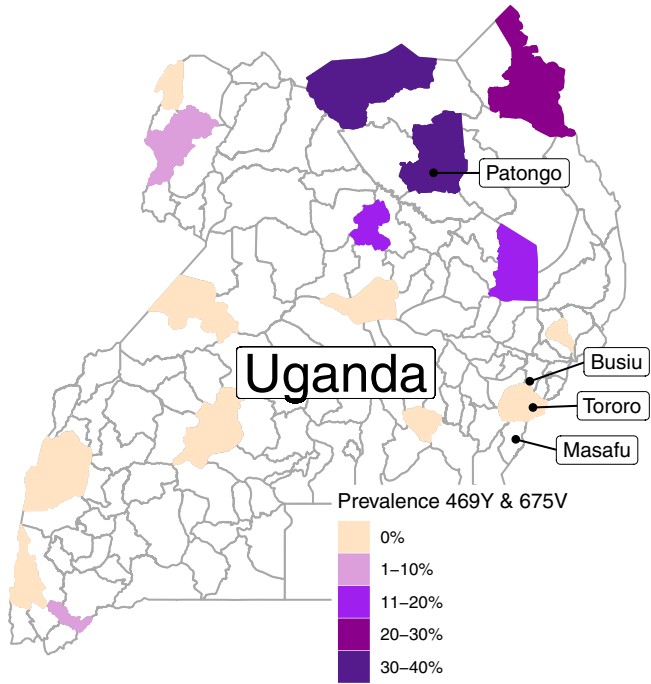

**Fig. 1 | Map of Uganda.** Study sites are labeled. The combined prevalence of the PfK13 469Y and 675V mutations in districts with available surveillance data from 2019[21] is shown with the color scale. Source data are provided as a Source Data file.

## Results

### Collection of *P. falciparum* isolates from northern and eastern Uganda

To compare isolates from different parts of the country, we assessed genotypes, growth inhibition, and ring survival for isolates collected from patients presenting with uncomplicated falciparum malaria at Patongo Health Center III, Agogo District, in northern Uganda (n = 57), and Tororo District Hospital, Tororo District and Busiu Health Center IV, Mbale District, two nearby sites in eastern Uganda (n = 42), from May 31 to August 16, 2021 (Fig. 1). Features of the participants providing these samples and characteristics of their infections are described in Table 1. In a broader survey, we assessed genotypes and growth inhibition for isolates from 819 patients presenting with uncomplicated falciparum malaria, including the sites noted above and Masafu Hospital, Busia District, from December 9, 2015 to August 20, 2021 (Table 1). Drug susceptibilities for the samples from eastern Uganda through 2019 were published previously[32]; in this report we focus on associations between in-depth sequencing of potential resistance mediators and drug susceptibility phenotypes.

### Genotypes of *P. falciparum* isolates from northern and eastern Uganda

For isolates collected concurrently in northern and eastern Uganda in 2021, we used molecular inversion probe (MIP) assays to characterize sequences of 80 genes potentially mediating drug resistance (Supplementary Table 1). Of particular interest were parasite polymorphisms recently associated with altered sensitivity to components of ACTs, namely PfK13 propeller domain mutations associated with partial resistance to artemisinins, and polymorphisms associated with altered sensitivity to lumefantrine (*pfmdr1* amplification and mutations) and piperaquine (*plasmepsin 2/3* amplification and novel PfCRT mutations). Consistent with recent data, prevalences of the PfK13 C469Y and A675V mutations were greater in northern, compared to eastern Uganda (C469Y 34% vs. 3%, p < 0.001; A675V 13% vs. 3%, p = 0.06, Table 2). Other PfK13 propeller domain mutations were uncommon; 15 mutations upstream of the propeller domain were

**Table 1 | Characteristics of participants providing *P. falciparum* isolates**

| Characteristic | Isolates with only IC$_{50}$ results | Isolates with RSA and/or IC$_{50}$ results | |
|---|---|---|---|
| Site | Eastern Uganda (*N* = 819) | Northern Uganda (*N* = 57) | Eastern Uganda (*N* = 42) |
| Dates of collection | Dec. 9, 2015–Aug. 20, 2021 | May 31–Aug. 17, 2021 | |
| Median age (years, (IQR)) | 3 (2–5) | 11 (4–14) | 5 (5–9) |
| Number male (%) | 366 (45) | 22 (39) | 21 (50) |
| Parasitemia (%, (IQR)) | 3.0 (1.7–5.0) | 1.0 (0.5–1.5) | 2.4 (1.7–3.6) |
| Median complexity of infection (IQR) | 2 (2–3) | 2 (2–3) | 2 (2–3) |
| Median time to assay (hours, (IQR)) | NA | 25.0 (24.3–26.0) | 24.0 (22.9–24.8) |

*NA* not available.

**Table 2 | Genetic polymorphisms in *P. falciparum* isolates from northern and eastern Uganda**

| Protein | Polymorphism | Prevalence | | *p* |
|---|---|---|---|---|
| | | Northern Uganda | Eastern Uganda | |
| PfK13 | C469Y | 18/53 (34%) | 1/39 (3%) | <0.001 |
| | A675V | 7/53 (13%) | 1/39 (3%) | 0.06 |
| | A578S | 1/53 (2%) | 1/39 (3%) | 1.0 |
| | G533A | 1/53 (2%) | 0/39 | 1.0 |
| PfCRT | K76T | 0/42 | 0/38 | 1.0 |
| PfMDR1 | N86Y | 0/48 | 0/39 | 1.0 |
| | Y184F | 34/48 (71%) | 31/39 (79%) | 0.46 |
| | D1246Y | 0/48 | 2/38 (5%) | 0.19 |
| | Copy number >1.5 | 0/37 | 0/34 | 1.0 |
| PfDHFR | N51I | 48/48 (100%) | 39/39 (100%) | 1.0 |
| | C59R | 46/48 (96%) | 38/39 (97%) | 1.0 |
| | S108N | 48/48 (100%) | 39/39 (100%) | 1.0 |
| | I164L | 2/47 (4%) | 4/39 (10%) | 0.40 |
| PfDHPS | A437G | 44/46 (96%) | 39/39 (100%) | 0.50 |
| | K540E | 47/48 (98%) | 39/39 (100%) | 1.0 |
| | A581G | 2/51 (4%) | 3/39 (8%) | 0.64 |
| PM2 | Copy number >1.5 | 0/33 | 0/31 | 1.0 |

Polymorphisms include all propeller domain (beginning at codon 440) variants in PfK13 and those known to be associated with drug susceptibility for the other proteins noted. Values shown are for combined mixed and pure mutant samples (with the mutant designation based on the 3D7 reference genome sequence). *P*-values were determined using two-sided Fisher's Exact tests, without corrections for multiple comparisons.

identified. Neither amplification of the *pfmdr1* or *plasmepsin 2/3* genes nor PfCRT mutations associated with piperaquine resistance in Asia (T93S, H97Y, F145I, I218F, M343L, G353V)[40,41] were seen. Considering other well-described polymorphisms previously associated with drug resistance, the prevalences of mutations in the transporters PfCRT (K76T) and PfMDR1 (N86Y, D1246Y) associated with altered sensitivity to aminoquinolines were very low, the prevalences of five mutations in PfDHFR (N51I, C59R, S108N) and PfDHPS (A437G, K540E) associated with resistance to the antifolates pyrimethamine and sulfadoxine were very high, and additional mutations in PfDHFR (I164L) and PfDHPS (A581G) associated with high level antifolate resistance were seen at low prevalences, all consistent with recent data from Uganda (Table 2)[20,21].

### Ex vivo drug susceptibilities of *P. falciparum* isolates from northern and eastern Uganda
For the isolates from 2021 studied with the ring survival assay (RSA), samples were stored at 4 °C and transported on the day of collection to our laboratory in Tororo. Time in storage before initiation of culture was slightly longer for samples collected in northern vs eastern Uganda (median 25.0 h in northern vs 24.0 h in eastern Uganda, *p* = 0.001). Susceptibilities to a panel of seven antimalarials were studied with a standard growth inhibition (IC$_{50}$) assay and, for DHA, with the ex vivo RSA. Assays for parasites collected in the two regions of the country were run in parallel. Susceptibilities for chloroquine, desethylamodiaquine (the active metabolite of amodiaquine), piperaquine, mefloquine, and pyronaridine were similar between the two sites and, for each of these drugs, IC$_{50}$s were at levels generally considered highly susceptible (Table 3). In contrast, for lumefantrine and DHA, isolates from northern Uganda were significantly less susceptible (after Bonferroni correction) than those from eastern Uganda (lumefantrine IC$_{50}$ 14.6 vs. 6.9 nM, *p* < 0.0001; dihydroartemisinin IC$_{50}$ 2.3 vs. 1.5 nM, *p* = 0.003; Table 3). For lumefantrine, 14 of 49 isolates from northern, but none of 38 from eastern Uganda had IC$_{50}$ >20 nM (*p* = 0.0002; Fig. 2). RSAs, which entailed counting parasites 72 h after initiation of a 6 h DHA pulse, did not show significant differences in survival between isolates from northern (median 2.5% survival for 52 samples) and eastern (median 1.4% survival for 29 samples; *p* = 0.53) Uganda, but of note some parasites with PfK13 mutations associated with delayed parasite clearance were present at both sites (Fig. 2).

### Associations between ring survival and *P. falciparum* genotypes
In the 81 samples from 2021 with available RSA data, we searched for associations between ring survival and PfK13 genotypes. Isolates with the PfK13 469Y mutation had significantly greater survival, with the survival highest for pure mutant (median 7.3%) compared to mixed (2.5%) and pure wild type (1.4%) parasites (Fig. 3). The PfK13 675 V mutation was not associated with increased survival, but this analysis was limited by sample size, with only 2 pure mutant 675 V isolates available for study. For broader consideration of potential resistance determinants, we used Mann–Whitney Wilcoxon tests to search for associations between polymorphisms in 23 genes for which variants have been associated with artemisinin partial resistance and our RSA results (Supplementary Table 1). Uncommon polymorphisms were assessed as aggregate variables, as described in Methods. Polymorphisms in five genes were associated with decreased or increased ring survival compared to wild type (Supplementary Table 2). Of these genes, considering the more prevalent allele (major allele) as wild type and the minor allele as mutant, any of 8 mutations in Pfcoronin (survival 6.5% in mutant vs. 2.3% in wild type, *p* = 0.04) or any of 17 mutations in falcipain 2a (survival 3.2% in mutant vs. 0.6% in wild type, *p* = 0.02) were most strongly associated with increased ring survival either alone or in association with PfK13 mutations (Fig. 4). Considering combined impacts, the presence of PfK13 469Y plus any mutation in Pfcoronin (median survival 12.6% in double mutant vs. 1.4% in double wild type, *p* = 0.009) or falcipain-2a (median survival 3.9% in double mutant vs. 0% in double wild type, *p* = 0.004) was also associated with increased survival. Of note, although some isolates with wild type PfK13 sequence had high levels of ring survival, these outliers appeared to be explained by mutations in falcipain-2a; all isolates with ring survival >5% had either the PfK13 469Y mutation, mutations in falcipain-2a, or both. Interestingly, mutations in PfK13 outside the propeller

**Table 3 | Drug susceptibilities of *P. falciparum* isolates from northern and eastern Uganda**

| Drug | Concentration range studied (nM) | Northern Uganda | | | Eastern Uganda | | | *p* |
|---|---|---|---|---|---|---|---|---|
| | | *n* | IC$_{50}$ (nM) | Range | *n* | IC$_{50}$ (nM) | Range | |
| LM | 0.05–1000 | 49 | 14.6 | 4.7–89.9 | 38 | 6.9 | 2.9–16.2 | *p* < 0.0001 |
| CQ | 0.5–10,000 | 49 | 11.0 | 5.4–250.0 | 38 | 10.1 | 6.1–31.8 | *p* = 0.16 |
| DAQ | 0.25–5000 | 49 | 9.3 | 3.4–33.0 | 38 | 7.7 | 2.7–17.7 | *p* = 0.03 |
| PQ | 0.05–1000 | 49 | 6.7 | 1.6–54.8 | 38 | 5.4 | 1.6–17.9 | *p* = 0.33 |
| MQ | 0.25–5000 | 49 | 12.1 | 2.2–38.6 | 38 | 8.7 | 1.4–24.7 | *p* = 0.01 |
| DHA | 0.05–1000 | 49 | 2.3 | 0.6–6.9 | 38 | 1.5 | 0.4–5.4 | *p* = 0.003 |
| PD | 0.05–1000 | 49 | 1.3 | 0.2–5.9 | 38 | 1.0 | 0.2–5.5 | *p* = 0.09 |

IC$_{50}$s are expressed as medians. *P*-values were determined using two-sided Mann–Whitney Wilcoxon tests, without corrections for multiple comparisons. With Bonferonni correction significance is at *p* = 0.007.

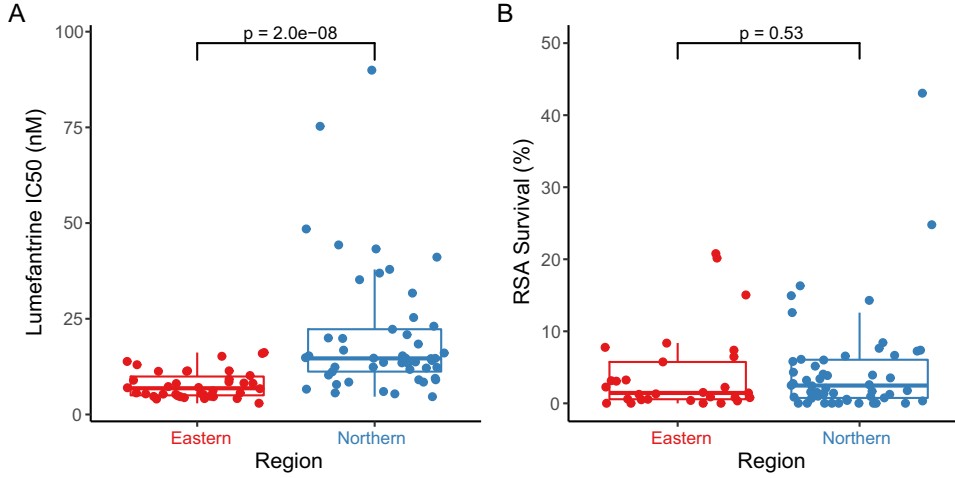

**Fig. 2 | Ex vivo susceptibilities.** Susceptibilities to lumefantrine (**A** growth inhibition assay) and DHA (**B** RSA) in parasites from northern and eastern Uganda. Each point represents the result for a single isolate (**A** *n* = 49 for northern and 38 for eastern Uganda; **B** *n* = 52 for northern and 29 for eastern Uganda). *P*-values were determined using the two-sided Mann–Whitney Wilcoxon test. Center bounds of boxes correspond to the median, and minimal and maximal bounds correspond to 25th and 75th percentiles, respectively. Whiskers extend to extreme values no further than 1.5x the IQR from the 25th or 75th percentiles. Source data are provided as a Source Data file.

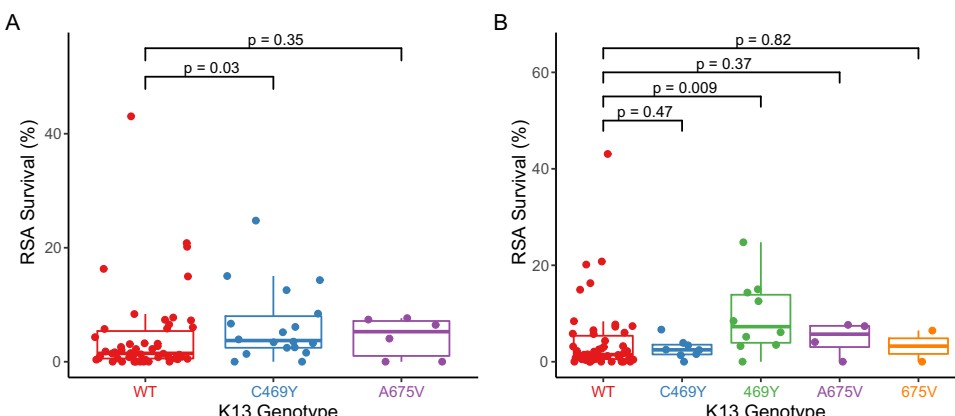

**Fig. 3 | Ex vivo ring survival in parasites with different PfK13 genotypes.** Each point represents the result for a single isolate (**A** *n* = 50 WT, 18 C469Y, 6 A675V; **B** *n* = 50 WT, 8 C469Y, 10 469Y, 4 A675V, 2 675V). Results are shown with mixed and pure mutant isolates combined (**A**) or shown separately (**B**). *P*-values were determined using the two-sided Mann–Whitney Wilcoxon test. Center bounds of boxes correspond to the median, and minimal and maximal bounds correspond to 25th and 75th percentiles, respectively. Whiskers extend to extreme values no further than 1.5x the IQR from the 25th or 75th percentiles. WT, wild type. Source data are provided as a Source Data file.

domain were associated with decreased RSA survival (the opposite of the effect of PfK13 469Y); considering any of 15 non-propeller mutations identified, survival was 1.5% in 42 mutant vs. 5.5% in 26 fully wild type isolates (*p* = 0.02; Supplementary Table 2). Two other polymorphisms with significant independent associations were deletion in a putative ubiquitin carboxyl-terminal hydrolase 1, associated with increased RSA survival, and a SNP in a gene encoding a protein of unknown function, associated with decreased survival.

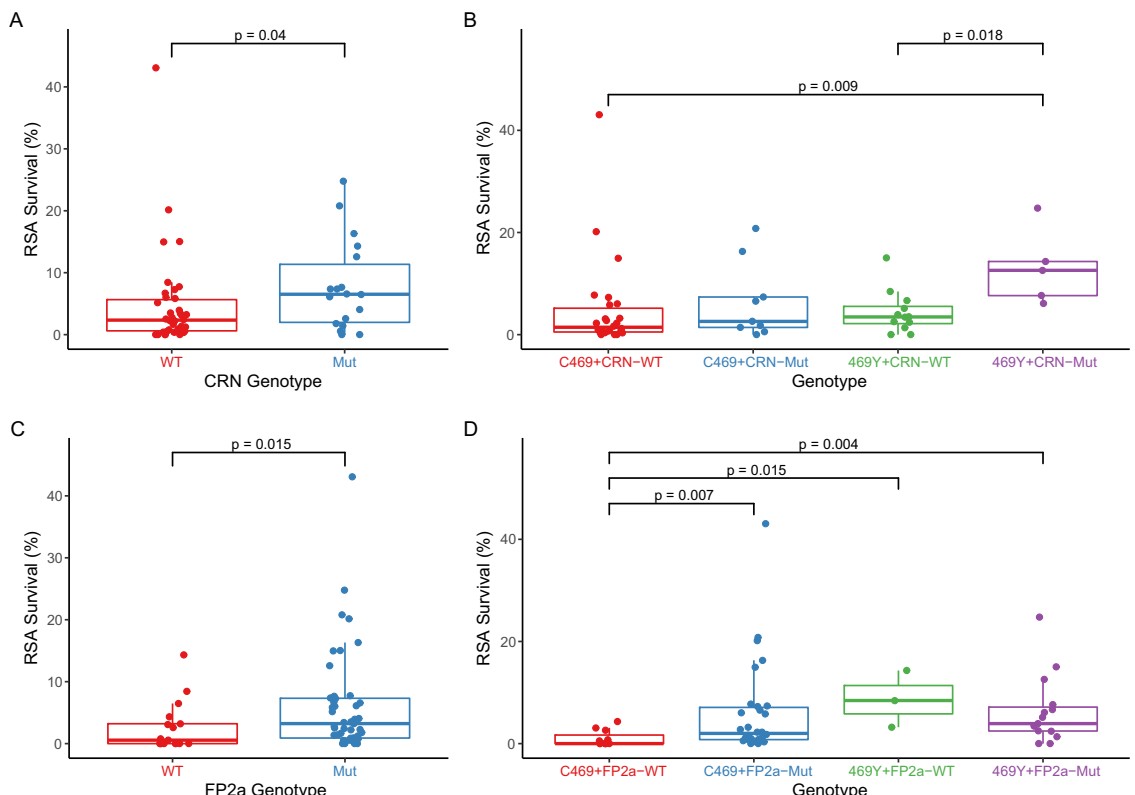

**Fig. 4 | Ex vivo susceptibilities in parasites with different genotypes.** Susceptibilities to DHA by RSA in parasites with different PfK13, Pfcoronin (CRN; **A**, **B**), and falcipain-2a (FP2a; **C**, **D**) genotypes are shown. Each point represents the result for a single isolate (**A** n = 38 WT, 18 Mut; **B** n = 26 C469 + CRN-WT, 9 C469 + CRN-Mut, 12 469Y + CRN-WT, 5 469Y + CRN-Mut; **C** n = 17 WT, 47 Mut; **D** n = 11 C469 + FP2a-WT, 30 C469 + FP2a-Mut, 3 469Y + FP2a -WT, 15 469Y + FP2a-Mut). PfK13 genotypes at codon 469 are indicated; mutant 469Y includes mixed isolates. P-values were

determined using the two-sided Mann–Whitney Wilcoxon test. Center bounds of boxes correspond to the median, and minimal and maximal bounds correspond to the 25th and 75th percentiles, respectively. Whiskers extend to extreme values no further than 1.5x the IQR from the 25th or 75th percentiles. WT, wild type, Mut, any Pfcoronin or falcipain 2a polymorphism. Source data are provided as a Source Data file.

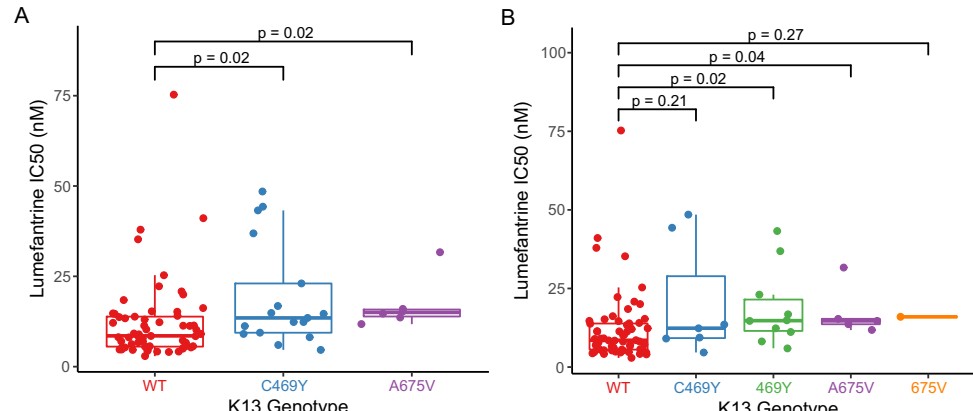

**Fig. 5 | Lumefantrine susceptibility of parasites with different PfK13 genotypes.** Each point represents the result for a single isolate (**A** n = 61 WT, 17 C469Y, 6 A675V; **B** n = 61 WT, 7 C469Y, 10 469Y, 5 A675V, 1 675V). Results are shown with mixed and pure mutant isolates combined (**A**) or shown separately (**B**). P-values were determined using the two sided Mann–Whitney Wilcoxon test. Center bounds of boxes

correspond to the median, and minimal and maximal bounds correspond to the 25th and 75th percentiles, respectively. Whiskers extend to extreme values no further than 1.5x the IQR from the 25th or 75th percentiles. WT, wild type. Source data are provided as a Source Data file.

## Associations between growth inhibition and *P. falciparum* genotypes

Comparing results for samples collected from northern and eastern Uganda in 2021, isolates with the PfK13 469Y or 675 V mutation were significantly less susceptible to lumefantrine (IC$_{50}$ 14.1 nM for mixed or mutant 469Y, 14.7 nM for mixed or mutant 675 V, and 8.7 nM for wild

type; Fig. 5). Susceptibilities to the other tested drugs, including mefloquine, which appears to share some determinants of decreased activity with lumefantrine[15], did not differ between wild type and PfK13 mutant parasites.

To more broadly consider associations between phenotypes and genotypes, we utilized a large set of 819 isolates with growth inhibition

**Table 4 | Polymorphisms associated with ex vivo lumefantrine susceptibilities**

| Gene | Polymorphism | Median LM IC$_{50}$ (nM, (N)) | | | | |
| --- | --- | --- | --- | --- | --- | --- |
| | | Wild type | Mixed | Mutant | Mixed/mutant | $p$ |
| K13 PF3D7_1343700 | C469Y | 5.1 (651) | 9.2 (14) | 14.8 (10) | 12.1 (24) | $6 \times 10^{-8}$ |
| | A675V | 5.2 (654) | 12.4 (9) | 8.9 (3) | 12.1 (12) | 0.001 |
| MDR1 PF3D7_0523000 | N86Y | 5.3 (642) | 2.6 (3) | 1.2 (2) | 1.7 (5) | 0.004 |
| | Y500N | 5.2 (505) | 14.7 (5) | 8.5 (1) | 13.2 (6) | 0.002 |
| | F938Y | 5.3 (610) | 4.0 (24) | 2.2 (5) | 3.9 (29) | 0.001 |
| Patatin-like phospholipase, putative PF3D7_0218600 | N1322_N1325dup | 5.5 (356) | 5.0 (218) | 3.5 (3) | 5.0 (221) | 0.004 |
| | I1478N    appears as haplotype N1480I N1481S | 5.6 (283) | 4.9 (264) | 5.3 (4) | 4.9 (268) | 0.002 |
| | G1494D | 5.2 (569) | 7.1 (19) | 20.0 (1) | 8.6 (20) | 0.006 |
| ApiAP2 PF3D7_0613800 | S256P | 5.3 (621) | 4.6 (55) | 3.7 (2) | 4.6 (57) | 0.002 |
| | N2135T | 5.0 (378) | 9.3 (5) | 11.0 (1) | 10.2 (6) | 0.01 |
| | H3564_D3569dup | 5.3 (635) | 2.8 (14) | 2.0 (1) | 2.8 (15) | 0.004 |
| | Y2222_N2223insKNY | 5.6 (300) | 4.9 (128) | 4.6 (55) | 4.7 (183) | 0.001 |
| | E205A | 6.5 (100) | 5.3 (94) | 4.3 (9) | 5.1 (103) | 0.007 |
| | D1149N | 5.3 (455) | 4.9 (172) | 4.0 (18) | 4.9 (190) | 0.001 |
| MRP1 PF3D7_0112200 | F334S | 5.0 (366) | 28.2 (2) | 13.0 (1) | 13.0 (3) | 0.01 |
| MRP2 PF3D7_1229100 | D630N | 5.6 (236) | 4.6 (100) | 5.1 (44) | 4.9 (144) | 0.003 |
| | N642D | 5.6 (259) | 4.5 (89) | 5.0 (32) | 4.8 (121) | 0.01 |
| FP2a PF3D7_1115300 | V147L | 4.9 (218) | 7.3 (30) | 7.0 (17) | 7.1 (47) | 0.001 |
| FP3 PF3D7_1115400 | A422T | 5.2 (601) | 1.4 (5) | 3.3 (1) | 1.8 (6) | 0.003 |
| PM1 PF3D7_1407900 | V20L | 5.1 (473) | 28.3 (2) | 16.8 (1) | 16.8 (3) | 0.008 |

Wild type and mutant designations are based on the 3D7 reference strain. P-values were determined using two-sided Mann–Whitney Wilcoxon tests, without corrections for multiple comparisons. $P \leq 0.01$ were considered significant.

data, collected mostly from eastern Uganda from 2015-2021, and also including 49 samples collected in northern Uganda in 2021, as described above. Parasite DNA was genotyped using the MIP platform targeting 80 genes encoding proteins potentially associated with drug susceptibility, including proteins known or predicted to be drug transporters and proteins for which polymorphisms are known or suspected of contributing to resistance to established antimalarials and compounds under development (Supplementary Table 1). We identified a total of 4337 polymorphisms and used Mann–Whitney Wilcoxon tests to explore relationships between variation in ex vivo IC$_{50}$s and the presence of these polymorphisms.

We first focused on lumefantrine, due to identification of significant differences in susceptibility in samples from northern and eastern Uganda, as described above. In evaluation of 713 isolates, we identified 33 non-synonymous polymorphisms associated with lumefantrine IC$_{50}$, with significance at $p \leq 0.01$. We then filtered to include only loci where median IC$_{50}$s for mixed and mutant alleles trended in the same direction (both had increased or decreased drug sensitivity relative to parasites with pure wild type genotypes) and for which the pure minority allele was present in at least one sample. This resulted in identification of polymorphisms in 9 genes associated with altered susceptibility to lumefantrine (Table 4). Most notably, the PfK13 mutations that have emerged in northern Uganda were strongly associated with decreased susceptibility to lumefantrine (469Y, $p = 6 \times 10^{-8}$; 675 V $p = 0.001$). Other associations, with either increased or decreased susceptibility to lumefantrine, included SNPs in PfMDR1, including the 86Y mutation which is now very uncommon, but was previously associated with increased lumefantrine susceptibility; and polymorphisms in a putative phospholipase; the ApiAP2 transcription

factor; the multidrug-resistance proteins PfMRP1 and PfMRP2; and the hemoglobinases falcipain-2a, falcipain-3, and plasmepsin I (Table 4).

Considering the other six tested drugs, applying the same filtering criteria as for lumefantrine, associations between a variety of polymorphisms and drug susceptibility were identified, mostly at relatively low levels of significance (Supplementary Tables 3–8). Most striking, there was strong association between well characterized mutations in PfCRT, including the K76T mutation known to mediate chloroquine resistance, and susceptibility to chloroquine ($p = 10^{-7}$–$10^{-20}$ for 6 PfCRT mutations) and the related drug monodesethylamodiaquine ($p = 0.01$–0.0003 for 4 PfCRT mutations). These results were expected[15], and they support the overall validity of our analyses.

## Stability of ex vivo lumefantrine susceptibility phenotypes

Anecdotally, some remarkable ex vivo drug susceptibility phenotypes have proven to be unstable, with IC$_{50}$s for antimalarial drugs lower after culture adaptation compared to values seen immediately after sample collection, potentially explained by more successful growth of drug sensitive compared to drug resistant strains in mixed cultures. To explore the stability of observed phenotypes we culture adapted northern Uganda strains with remarkable lumefantrine IC$_{50}$ results. Lumefantrine susceptibility was generally greater in culture adapted parasites compared to initial ex vivo values. For eight isolates with original ex vivo lumefantrine IC$_{50}$ values >30 nM (median IC$_{50}$ 39.5 nM), subsequent IC$_{50}$ values measured after culture for 4 or more weeks and then after freezing, thawing, growth in culture, and repeat assays, were generally lower than the initial ex vivo values (Table 5). However, after culture adaptation IC$_{50}$ values for these eight northern Uganda isolates measured before (median IC$_{50}$ 13.2 nM) or after

**Table 5 | Lumefantrine susceptibilities before and after culture adaptation**

| Isolate | Ex vivo | After culture adaptation | After culture adaptation, freeze/thaw, and reculture | |
|---|---|---|---|---|
| | | | Uganda | USA |
| PAT-015 | 35.2 | 10.7 | 18.2 | 5.6 |
| PAT-023 | 37.9 | 38.0 | 22.7 | 10.1 |
| PAT-026 | 36.9 | 11.5 | 15.7 | 11.4 |
| PAT-027 | 44.3 | 11.9 | 9.5 | 9.4 |
| PAT-033 | 89.9 | 42.7 | 18.7 | 7.4 |
| PAT-037 | 43.3 | 14.3 | 4.8 | M |
| PAT-038 | 41.1 | 13.2 | 7.4 | 5.7 |
| PAT-045 | 31.7 | M | 24.4 | 5.0 |
| All isolates (median; IQR) | 39.5 (35.7–44.1) | 13.2 (11.5–38.0) | 16.9 (7.9–21.7) | 7.4 (5.6–10.1) |

Results shown are $IC_{50}$s (nM). M = missing. Identical assays on samples after culture adaptation, freeze/thaw, and reculture were performed in Uganda and the USA.

(median $IC_{50}$ 16.9 nM assessed in Uganda and 7.4 nM assessed in the USA) freeze-thaw-reculture were above values for parasites collected from eastern Uganda for direct comparison with the northern Uganda isolates (median $IC_{50}$ 6.9 nM for 38 isolates) or for a larger set of isolates published earlier (median $IC_{50}$ 5.1 nM for 365 isolates[32]).

## Discussion

The emergence in northern Uganda of *P. falciparum* harboring the PfK13 C469Y or A675V mutations raises concern regarding selection of parasites resistant to both artemisinins and ACT partner drugs, potentially leading to the inability of first-line ACTs to effectively treat malaria. To better appreciate the current situation, we directly compared the drug susceptibilities of isolates collected in 2021 from malaria patients in northern Uganda, where the PfK13 mutations have been common in recent years, and eastern Uganda, where they have been uncommon. We found, as expected, high prevalence of the C469Y and A675V mutations in isolates from northern Uganda, although prevalence was also higher than previously seen in isolates from eastern Uganda[21]. Using the DHA RSA, the standard measure of in vitro susceptibility to artemisinins, we saw an association between the C469Y mutation, but not the A675V mutation (for which few samples were available for study), and decreased parasite survival. Using standard growth inhibition assays, susceptibility to lumefantrine and DHA, but not other tested drugs, was significantly lower in isolates from northern, compared to eastern Uganda. Considering a much larger number of isolates collected from 2015–21, but for which only growth inhibition assays were available, decreased susceptibility to lumefantrine was strongly associated with the PfK13 469Y mutation. These results suggest that emergence of partial resistance to artemisinins has been accompanied by decreased activity of lumefantrine, potentially foretelling loss of efficacy of artemether-lumefantrine, the first-line antimalarial in Uganda and most of Africa.

It is important to determine if the PfK13 mutations recently identified in northern Uganda[19–22] are associated with increased parasite survival after exposure to artemisinins. Previous work identified association between the 675V mutation and clinical delayed clearance after treatment with artemisinins in southeast Asia[7]. In a recent study from northern Uganda both mutations were associated with clinical delayed clearance and only the 675V mutation with ex vivo increased survival after DHA exposure, but analyses were limited by relatively few mutant isolates for study[22]. A study from Rwanda showed that single isolates collected in 2019 with the 469 F, 561H, and 675 V mutations had in vitro increased survival compared to a wild type parasite[18], but to our knowledge the 469Y mutation had not previously

been associated with increased survival in an RSA. In our studies, the 469Y mutation was significantly associated with enhanced survival by RSA. Therefore, our results solidify the conclusion that both PfK13 mutations that emerged recently in northern Uganda are associated with artemisinin partial resistance.

Polymorphisms in a number of *P. falciparum* proteins have been shown to be associated with artemisinin partial resistance, either in conjunction with PfK13 mutations or independently[8,10], but consistent mediators other than PfK13 have not been identified, and mediation of resistance appears to be highly dependent on parasite genetic background[42]. Our evaluation of associations between polymorphisms in 23 candidate genes and enhanced survival in the RSA identified a number of polymorphisms with modest association with enhanced survival (Supplementary Table 2), and the strongest association with polymorphisms in two genes, encoding falcipain-2a and Pfcoronin.

Strikingly, all isolates with RSA survival >5% despite lack of the PfK13 469Y mutation had mutations in falcipain-2a. Of the 17 mutations identified in falcipain-2a, 11 were also identified in isolates from southeast Asia in which falcipain-2a haplotypes were associated with enhanced RSA survival[9]. Falcipain-2a is a hemoglobinase that plays a key role, in concert with other proteases, in hydrolyzing hemoglobin to supply amino acids for parasite metabolism[43]. Loss of falcipain activity due to treatment with specific inhibitors or gene knockout markedly blunted the antimalarial activity of DHA, indicating that falcipain-mediated proteolysis of hemoglobin is needed for efficient activation of artemisinins[44]. These results are consistent with our understanding that the conversion of artemisinins by heme, after liberation from hemoglobin, into toxic free radicals, is a prerequisite to efficient action of artemisinins[45]. Interestingly, a mutation encoding a falcipain-2a stop codon was identified in parasites selected in vitro for resistance to artemisinin, but this selection occurred after emergence of a PfK13 mutation[6]. Taken together, available data suggest that certain mutations in falcipain-2a can mediate artemisinin partial resistance independent of or in concert with mutations in PfK13.

In studies of parasites from Senegal, in vitro selection of *P. falciparum* with increasing concentrations of DHA selected for parasites with enhanced survival associated with mutations in Pfcoronin, but not PfK13[10]; impacts of Pfcoronin mutations on survival after DHA exposure varied based on parasite genetic background[46]. Pfcoronin is a member of the WD40-propeller domain protein family that is known to associate with actin filaments and intracellular membranes[47,48]. The biological basis of contributions of Pfcoronin mutations to artemisinin partial resistance is unknown, but it was noted that impacts of Pfcoronin mutations are masked by PfK13 mutations, suggesting that mutations in the two proteins may impact on the same parasite mechanisms[46].

Additional polymorphisms were associated with altered RSA survival. A deletion in *pfubp1*, which encodes a putative deubiquitinating enzyme, was associated with increased ring survival. Two mutations in the *P. chabaudi* homolog of this protein were associated with artemisinin resistance in a murine genetic cross[49]; mutations were linked to artemisinin partial resistance in *P. falciparum* in vitro[50] and *P. berghei* in mice[51], but not to clinical partial resistance[52]; and the protein was shown by affinity labeling to associate with PfK13[53]. Our results support these limited data identifying PfUBP1 mutations as potential secondary mediators of artemisinin partial resistance. Interestingly, the presence of any of 15 pre-propeller domain PfK13 mutations was associated with decreased survival. These results suggest that, perhaps paradoxically, these mutations enhance artemisinin activity, the opposite effect of propeller domain mutations. A polymorphism in a protein of unknown function (Pf3D7_1433800) was also associated with increased survival. A different polymorphism in this protein was selected in the same experiments that identified Pfcoronin as a potential resistance determinant[10], but this did not appear to contribute to increased ring survival[46].

We also compared susceptibility of *P. falciparum* isolates to a panel of seven antimalarials using a standard growth inhibition assay. Considering only isolates collected and studied over the same time frame in 2021, susceptibilities to two drugs, lumefantrine and DHA, were lower in isolates from northern Uganda than those from eastern Uganda. Considering a much larger set of isolates collected in 2015–21, polymorphisms in nine genes were associated with altered lumefantrine susceptibility compared to that of wild type parasites. These results highlight potential mediators of, or factors associated with, altered lumefantrine susceptibility. First, the PfK13 469Y and 675 V mutations were strongly associated with decreased susceptibility, highlighting the potential for partial resistance to artemisinins to be associated with decreased susceptibility to partner drugs. However, these results do not indicate a causal role for PfK13 mutations in altered lumefantrine susceptibility. Second, three mutations in the putative drug transporter PfMDR1 were associated with altered lumefantrine susceptibility; one of these, 86Y, was previously at high prevalence in Africa, was selected against by exposure to lumefantrine[15], and was associated with increased drug susceptibility in prior studies[31,32] and our current analysis. Third, mutations in a putative phospholipase were associated with altered lumefantrine susceptibility. Mutations in this protein were previously selected by in vitro exposure to primaquine and associated with decreased susceptibility to that drug[54]. Interestingly, loss-of-function mutations in another predicted *P. falciparum* phospholipase, PfPARE, were associated with decreased susceptibility to MMV011438, a series of pepstatin esters[55], and the antimalarial candidate oxoborole AN13762[56], presumably due to loss of intracellular drug activation. Fourth, different polymorphisms in a predicted *P. falciparum* transcription factor, ApiAP2[57], were associated with both increased and decreased lumefantrine susceptibility. Different mutations in this protein were associated with decreased susceptibility to three different compounds under development as antimalarials in a chemogenetic screen[54] and to quinine in a genome-wide association study[58]. Fifth, mutations in the putative drug transporters PfMRP1 and PfMRP2 were associated with alterations in lumefantrine susceptibility. Other mutations in PfMRP1 have been described in African parasites and associated with prior therapy with artemether-lumefantrine[59] and with decreased susceptibility to chloroquine, artemisinin, mefloquine, lumefantrine, piperaquine, and/or DHA in different studies[60–63]. A deletion in the PfMRP2 upstream promoter[64] and a number of polymorphisms in the coding sequence[65] were associated with decreased susceptibility to aminoquinolines. Sixth, SNPs in genes encoding hemoglobinases, the cysteine proteases falcipain-2a and falcipain-3[43] and the aspartic protease plasmepsin-I[66], were associated with altered lumefantrine activity. As noted above, decreased activity of falcipain-2a through enzyme inhibition or gene knockout caused decreased susceptibility to artemisinins[44], and altered function of other hemoglobinases might also impact on artemisinin action.

The identification of parasites with unusually high lumefantrine IC$_{50}$s from northern Uganda led to interest in characterizing this phenotype. However, the phenotype proved to be unstable. Growth of parasites in culture for 4 or more weeks to establish culture adapted parasites was accompanied by an increase in susceptibility to lumefantrine. Thus, it appears that some parasites with decreased lumefantrine sensitivity are circulating in northern Uganda, but that, in the mixed isolates typically present in this region, the less susceptible strains are routinely out-competed by more drug sensitive strains during culture, and/or decreased susceptibility phenotypes are lost in vitro due to unexplained environmental differences between in vivo and in vitro growth of parasites.

Our study had some important limitations. First, due to the logistical challenges of studying isolates from Agago District in our laboratory in Tororo District (~400 km distant), we collected parasites for RSAs only over a 2-month period, limiting our analysis to 57 isolates from northern Uganda. Second, due to our interest in studying isolates collected during the same time interval, we were limited to analysis of RSAs for 42 isolates from eastern Uganda. The limited sample size limited the power of our RSA analyses. Third, participants who provided samples from northern and eastern Uganda differed in some respects, notably lower age and higher parasitemia in those from eastern Uganda; however, it is not expected that these differences would impact on ex vivo measures of drug susceptibility. Fourth, isolates were usually polyclonal, as typical for infections in high transmission regions of Africa. Ex vivo IC$_{50}$s necessarily represented averages of activities of clones in a sample, and genomic analyses may have missed some sequences of minority clones. Fifth, our multiplex deep sequencing approach limited analysis of portions of some genes due to incomplete tiling or inadequately balanced PCR conditions. Sixth, our studies were limited to the target genes and not able to characterize the full complement of variation across the genome in relation to drug susceptibility; additional studies, including genome-wide association studies and genetic crosses, will be helpful in this regard. Despite these limitations, we feel that the robust associations identified are valid, but additional study of larger numbers of isolates collected across a wide geographic range are certainly a high priority.

The clinical ramifications of decreased activity of both artemisinins and lumefantrine against *P. falciparum* circulating in northern Uganda are unknown. Clearance of parasites with PfK13 469Y and 675 V mutations was delayed, compared to that of wild type parasites, after treatment with intravenous artesunate[22]. This result suggests that responses to treatment of severe malaria caused by mutant parasites may be slow, leading to increased severe morbidity and mortality. In Rwanda, clearance of parasites with a different PfK13 mutation, 561H, was delayed compared to that of wild type parasites, after treatment with artemether-lumefantrine, but treatment efficacy assessed by standard measures did not differ between patients infected with mutant and wild type parasites[17]. Impacts of the Ugandan 469Y and 675 V mutations on the treatment efficacies of ACTs are unknown. However, altered responses to both artemisinins and lumefantrine, as suggested by our data, appear likely to affect responses to therapy with artemether-lumefantrine, the first-line antimalarial in Uganda and most malaria-endemic countries in Africa. Studies in regions with PfK13 mutant parasites of the antimalarial efficacies of intravenous artesunate to treat severe malaria and of artemether-lumefantrine to treat uncomplicated malaria are thus of the highest priority.

## Methods

### Collection of *P. falciparum* isolates

For assessments that included RSAs and growth inhibition assays, samples were obtained from May 31–August 17, 2021 from 3 sites: Patongo Health Center III, Agogo District, in northern Uganda; Tororo District Hospital, Tororo District, in eastern Uganda; and Busiu Health Center IV, Mbale District, also in eastern Uganda (Fig. 1). For assessments that included only growth inhibition assays, samples were obtained from December, 2015-August, 2021 from the sites in eastern Uganda noted above, as well as Masafu Hospital, Busia District, also in eastern Uganda, as described previously[32]. For Patongo Health Center, samples were collected twice weekly and transported on the day of collection to our laboratory in Tororo. Samples from eastern Uganda sites were collected as available on a daily basis. Isolates were collected from patients aged 6 months or older presenting at the sites with clinical symptoms of malaria, a positive Giemsa-stained blood film for *P. falciparum*, and no signs of severe disease. Patients reporting use of antimalarial treatment within the previous 30 days or with evidence of an infection with other *Plasmodium* species were excluded. Written informed consent was obtained from all participants. Parents or guardians of children younger than 18 years provided written consent on their behalf; children aged 8–17 years provided assent. 2–5 mL of venous blood was collected in a heparin tube before the start of

therapy. Participants were administered artemether-lumefantrine, following national guidelines, after sample collection. The studies were approved by the Makerere University Research and Ethics Committee, the Uganda National Council for Science and Technology, and the University of California, San Francisco Committee on Human Research.

## Parasite culture and collection of samples for DNA analysis

Parasitemia was identified with Giemsa-stained thin films using a light microscope with a 100× objective lens and counting 1000 or more erythrocytes. Due to logistical considerations (samples collected at different times of the day and from different locations) samples were stored at 4 °C, with culture generally initiated on the morning after sample collection. Parasites were placed in culture as previously described[32]. Blood was centrifuged for 10 min at room temperature, plasma and buffy coat were removed, and the erythrocyte pellet was washed three times with RPMI 1640 media (Thermo Fisher Scientific, Waltham, MA, USA) at 37 °C. The pellet was resuspended in complete medium consisting of RPMI 1640 with 25 mM HEPES, 24 mM $NaHCO_3$, 0.1 mM hypoxanthine, 10 μg/mL gentamicin, and 0.5% AlbuMAX II (Thermo Fisher Scientific, Waltham, MA, USA) to produce a hematocrit of 50%. In addition, 4 aliquots of the washed pellets of approximately 10 μL were spotted onto Whatman 3MM filter paper (Cytivia, Marlborough, MA, USA) for subsequent molecular analysis.

## Ex vivo growth inhibition assays

Drug susceptibilities were evaluated in samples with a minimum of 0.3% parasitemia using a 72 h microplate growth inhibition assay with SYBR Green detection, as previously described[32]. Study compounds (chloroquine, monodesethylamodiaquine, piperaquine, lumefantrine, mefloquine, dihydroartemisinin, and pyronaridine), supplied by Medicines for Malaria Venture, were dissolved in dimethyl sulfoxide (except distilled water for chloroquine) as 10 mM stocks and stored at −20 °C. Drugs were serially diluted by a factor of 3 in complete medium in 96-well microplates (50 μL per well), including drug-free and parasite-free control wells, with concentrations optimized to capture full dose-response curves. Cultures were diluted with uninfected erythrocytes from local blood banks for total volumes of 200 μL per well at 0.2% parasitemia and 2% hematocrit. Plates were maintained at 5% $CO_2$, 5% $O_2$, and 90% $N_2$ for 72 h at 37 °C in a humidified modular incubator (Billups Rothenberg, San Diego, CA, USA). After 72 h, well contents were resuspended and 100 μL culture per well was transferred to black 96-well plates containing 100 μL per well SYBR Green lysis buffer (20 mM Tris, 5 mM EDTA, 0.008% saponin, 0.08% Triton X-100, and 0.2 μL/mL SYBR Green I (Invitrogen, Thermo Fisher Scientific, Waltham, MA, USA)), and mixed. Plates were incubated for 1 h in the dark at room temperature, and fluorescence was then measured with a FLUOstar Omega plate reader (BMG LabTech, Cary, NC, USA; 485 nm excitation and 530 nm emission). To monitor stability of drug stocks, laboratory control *P. falciparum* Dd2 (MRA-156) and 3D7 (MRA-102) strains (BEI Resources, Manassas, VA, USA) were maintained in culture and assayed (beginning at the ring stage) approximately monthly, following synchronization with a magnetic column (Miltenyi Biotec, Auburn, CA, USA). For adaptation of parasites for long-term culture, freshly collected isolates were diluted to 1% parasitemia, if needed, using donor erythrocytes and cultured at 2% hematocrit in RPMI complete medium under the conditions described above. Cultures were diluted with uninfected erythrocytes as needed. After ~4 weeks, ring-stage parasites were cryopreserved in Glycerolyte-57 solution (Frensenius Kabi AG, Hamburg, Germany) and stored in gas-phase liquid nitrogen.

## Ex vivo ring survival assays

Standard growth inhibition assays do not predict clinical delayed clearance associated with artemisinin-resistant parasites. Therefore, DHA susceptibility was measured using the ex vivo RSA, as previously described[67]. In brief, freshly cultured parasites with a minimum of 0.2% parasitemia, prepared as described above for growth inhibition assays, were incubated with 700 nM DHA or, for controls, 0.1% DMSO, for 6 h. Isolates with >1% parasitemia were diluted to 1%; isolates with ≤1% parasitemia were cultured at starting parasitemias. The cells were washed to remove drug after 6 h, and culture was continued for an additional 66 h. At 72 h after culture initiation, Giemsa-stained thin smears were prepared. Cultures were considered viable and appropriate for assessment if control parasitemia was ≥0.2% and also ≥25% of the starting parasitemia. Parasitemias were counted in control and treated cultures, and RSA survival was expressed as the proportion of viable parasites in the DHA treated cultures relative to controls.

## Genomic characterization of isolates

Gene sequences and copy number were analyzed by MIP capture and deep sequencing, as previously described[21,32,68]. For MIP capture, we targeted a total of 80 genes, selected because of known or potential roles in altered susceptibility to standard antimalarials or compounds under development (Supplementary Table 1), using previously published and newly designed probes (Supplementary Table 9). Novel probes were designed using MIPTools software (version 0.19.12.13). DNA was isolated with Chelex-100 extraction buffer as previously reported, using 0.01% Tween 20 instead of saponin[21]. MIP capture, library preparation, and sequencing were done as previously described[68]. Sequencing reads are available in the National Center for Biotechnology Information (NCBI) Sequence Read Archive (accession number PRJNA850445). MIPTools was used to organize raw sequencing data and to perform variant calling. Individual genotypes were assigned for polymorphic sites that were covered by a minimum of 10 unique molecular identifiers (UMIs) and variants were required to have a within sample allele count ≥3 UMIs for alternate alleles and ≥2 UMIs for reference alleles. Copy numbers were estimated on the basis of sample and probe normalized depth of sequence coverage from 31 unique probes for *pfmdr1* and 21 probes for *plasmepsin 2/3*. The Dd2 strain, which contains amplified *pfmdr1* and is single copy for *plasmepsin 2/3*, and the G8 subclone of KH001_053 (kindly provided by Selina Bopp and the TRAC collaboration), which has multiple copies of *plasmepsin 2*[69], were used as controls. Complexity of infection (COI) was estimated from genotyping data generated by MIP sequencing using THE REAL McCOIL, a Markov chain Monte Carlo model that estimates the allele frequency and COI of samples[70].

## Statistical analysis

$IC_{50}$ values were derived by plotting fluorescence intensity against log drug concentration and fit to a non-linear curve using a four-parameter Hill equation in Prism (version 9.0), as previously described[32]. All additional analyses were done in R (version 4.1.2). Categorical data were evaluated using two-sided Fisher's Exact tests and continuous data, including growth inhibition assays and RSAs, with two-sided Mann–Whitney Wilcoxon tests. For evaluation of $IC_{50}$ data, genotypes were considered as binary variables, determined by the presence or absence of the minor allele (mixed and pure minor allele genotypes were combined). $P \leq 0.01$ were considered significant. For RSA survival, loci with high (>15%) prevalence of minor alleles were evaluated independently, as described above, and loci with low (≤15%) minor allele prevalence were analyzed as aggregate variables, in which the presence of any minor allele was compared with the major allele. For both high and low prevalence loci, $p \leq 0.05$ were considered significant. For loci that were statistically significant, RSA survival was also evaluated in the context of K13 C469Y and A675V mutations. Candidate loci and K13 genotypes were treated as binary variables as described above.

**Reporting summary**

Further information on research design is available in the Nature Research Reporting Summary linked to this article.

## Data availability

All relevant data are available from the corresponding authors upon request. Sequence data are available in the National Center for Biotechnology Information (NCBI) Sequence Read Archive (accession number PRJNA850445). Data and code are available at https://github.com/PJRosenthalLab/2022_Tumwebaze_NatCom. Source data are provided with this paper.

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

## Acknowledgements

The study was funded by the National Institutes of Health (AI075045 (P.J.R.), AI089674 (P.J.R.), TW007375 (P.J.R.), and AI139520 (J.A.B.)) and the Medicines for Malaria Venture (RD/15/0001 (P.J.R.)). We thank study participants and staff members of the clinics where samples were collected. We thank Selina Bopp, Sarah Volkman and the members of the TRAC collaboration for kindly providing a KH001_053 clone as a copy number control.

## Author contributions

All authors conceived and designed the experiments; P.K.T., M.D.C., M.O., S.O., O.B., T.K., J.L, S.G., D.G., S.R.S., F.G.C., and R.A.C. acquired the data; P.K.T., M.D.C., M.O., S.O., T.K., S.G., D.G., J.A.B., R.A.C., and P.J.R. analysed and interpreted the data; P.K.T., M.D.C., S.L.N., J.A.B, R.A.C., and P.J.R. played primary roles in writing the manuscript. All authors approved the final manuscript.

## Competing interests

The authors declare no competing interests.
