## [Peer review file · Nature Communications]

REVIEWER COMMENTS

Reviewer #1 (Remarks to the Author):

NCOMMS-21-26394A-Z

'Decreased susceptibility of Plasmodium falciparum to both dihydroartemisinin and lumefantrine in northern Uganda'

Corresponding author: Melissa Conrad & Philip J. Rosenthal

Reviewer comments:

A. Summary of the key results

This manuscript submitted by Patrick K. Tumwebaze et al reports some important findings regarding the evolution of in vitro susceptibility of Ugandan *P. falciparum* parasites to lumefantrine, the companion drug used in combination with artemisinin (AL).

Using a simple but robust design, the authors investigated (including in vitro phenotypes and genotypes) *P. falciparum* isolates from Northern Uganda where artemisinin partial resistance is firmly established and from eastern Uganda, an area still free from artemisinin partial resistance.

The key results are:

- Genotyping: They confirm high proportions of validated Pfk13 mutants (469Y & 675V) in Northern Uganda compared to Eastern Uganda, the absence of parasites with pfmdr1 or Pfpm2/3 multicopy or mutations in Pfcrt gene previously associated with piperaquine resistance.

- In vitro phenotyping

- o Ex vivo RSA/ As observed for Pfk13 mutants, the median survival rate of samples from Northern U was higher than samples from eastern U.

- o IC50 assays: Susceptibilities to a panel of 7 antimalarials were similar between samples from Northern and Eastern U except for Lumefantrine and DHA (lower susceptibilities for Northern).

- Association between genotypes and phenotypes. This part is the most interesting and reveals:

- o Robust association of high RSA survival rates with Pfk13 469Y, Pfcoronin and Pffalcipain 2a mutants. These data are quite novel and confirm that several determinants (major and minor) are involved in partial artemisinin resistance.

- o Strong association between Lumefantrine IC50 and polymorphisms in 9 genes (based on a large set of data from samples collected in 2015-2021), including Pfk13, Pfmdr1, ApiAP2, Pfmrp1/2, falcipain 2/3

and plasmepsin 1 mutants. Again, these findings confirm the dynamic evolution and adaptation of *P. falciparum* parasites under ACT pressure.

- Generation of culture adapted parasites with various in vitro susceptibility phenotype to lumefantrine. Undoubtedly, these parasite lines constitute valuable materials for seeking molecular signatures associated with lumefantrine resistance through NGS analyses or genetic cross studies.

B. Originality and significance: if not novel, please include reference

Data presented in the manuscript are relevant and assess the current epidemiological situation regarding antimalarial drug resistance in Uganda. As Uganda has recently seen the emergence of artemisinin partial resistance, our efforts must be focused the surveillance of the resistance of *P. falciparum* parasites to partner drugs used in ACT and particularly lumefantrine. By using state of the art methods (phenotyping and genotyping), this study shows a stringly example of what must be done now in Africa.

Basically, this study clearly demonstrates the powerful of longitudinal surveys and cohort studies and the need to implement such approaches in Africa

C. Data & methodology: validity of approach, quality of data, quality of presentation

Major comment.

I have no major comment, only minor comments are presented below.

Minor comments.

Abstract

- Please replace 'artemisinin resistance' by 'artemisinin partial resistance' that is the official term, in the abstract and the main text.

- Last sentence. I suggest adding at the end of the sentence 'in Northern Uganda'.

Introduction

- Third sentence. I suggest changing the sentence as 'The resistance phenotype entails ex vivo delayed clearance of parasites after treatment with artemisinins, or in vitro prolonged survival of cultured parasites after exposure to dihydroartemisinin (DHA)'.

- Page 3, second paragraph, first sentence. I suggest to delete 'amodiaquine'. Amodiaquine resistance in Cambodia is likely linked to the use of chloroquine than artemisinin and I suspect that amodiaquine resistance have emerged long before the introduction of artemisinin.

- Page 4. Ref 35. I suggest adding a recent publication (Kubota et al, Front Cell Inf Microb, 2022).

Results

- Page 7, second paragraph, third sentence. The term 'delayed clearance' for in vitro assay is confusing. I suggest to replace by 'increase survival rate'.

Discussion

- Page 11, second paragraph, last sentence. I suggest adding 'in vitro' before 'artemisinin partial resistance'.

- Page 12, second paragraph. I suggest adding that mutation in Pffalcipain 2a has also been observed in the generation of the F32ART strain (Ariey et al, Nature, 2014). This codon stop occurred sequentially after the Pfk13 476I mutation (S69stop).

Methods

- Ex vivo RSA. Could you mention in the manuscript which reference strains were used as controls.

- Genomic characterization of isolates. Did you use a strain with multiple Pfp2m2 copy number as control for Pfp2m2 CNV assessment?

D. Appropriate use of statistics and treatment of uncertainties

The use of statistics and treatment of uncertainties are appropriate.

E. Conclusions: robustness, validity, reliability

The conclusions are reliable and cautious. The last paragraph clearly lists the limitations of the study.

F. Suggested improvements: experiments, data for possible revision

See earlier comments

G. References: appropriate credit to previous work?

Yes.

H. Clarity and context: lucidity of abstract/summary, appropriateness of abstract, introduction and conclusions

See comments above

Reviewer #2 (Remarks to the Author):

Tumwebaze and colleagues present important new data on drug susceptibility of the malaria parasite *Plasmodium falciparum* in Uganda, comparing an area with evidence of emerging resistance to a less affected area. They leverage their substantial knowledge, data and sample banks from previous studies to provide a rich knowledge-context in which to evaluate susceptibility phenotypes of recently collected clinical isolates. This is extremely valuable work and will have an immediate impact in the field, not least because it provides ample intelligence to support planning of new field, clinical and biological studies of this emerging resistance.

Noteworthy results include:

- thorough phenotypic studies ex vivo and in vitro that greatly solidify the evidence that the mutations emerging in Uganda, particularly C469Y, are associated with reduced artemisinin susceptibility
- evidence of co-selection of polymorphisms in other loci of interest - this includes important validation of coronin and UBP-1 as players in this system
- evidence that lumefantrine susceptibility is being eroded in parasites carrying K13 mutations. However the paper does not directly address the genetic basis of this.

MAJOR COMMENT

The authors target-based approach using MIP genotyping is sound, extensive and provided a wealth of data from a relatively restricted sample set. However only agnostic approaches (genetic crosses OR whole-genome shotgun sequencing and GWAS - requiring larger sample sets) can identify the full

complexity of the genetics underpinning both the DHA and lumefantrine susceptibility phenotypes. This needs to be clearly stated.

ABSTRACT

- second last sentence: the second phrase is difficult to follow. I suggest:

"...associated with altered drug response, notably PfK13 469Y with decreased susceptibility to lumefantrine ($p=6 \times 10^{-8}$) and PfCRT mutations with chloroquine resistance ($p=1 \times 10^{-20}$)."

- last sentence: unusually, you have understated the impact of your findings here. "raise concern"? Something braver is justified, such as

"Our results provide evidence of an emerging threat to artemether-lumefantrine therapeutic efficacy in Uganda."

INTRODUCTION

- second paragraph: in my view the adj. "massive" is too hyperbolic for a scientific treatise (but not as bad as "huge"). Please substitute with something more precise or appropriate.

- third paragraph: yes, 86Y was directly selected in Africa by AQ use but probably more extensive was its indirect selection by CQ, in both cases together with the CVIET haplotype of pfcr. Please try and add this if word count allows.

RESULTS

- practice in the field is to favour "EC50" over "IC50" currently. If there is a strong rationale for the latter I did not find it (assuming the assays are standard). Can you rule out cytotoxic drug effects as contributing to the apparent inhibition? A brief word in Methods section is justified if so.

- p5: did the authors find any evidence that a single *P. falciparum* isolate could harbour two or more K13 propeller mutations? Were C469Y and A675V ever seen together?

- p6 bottom paragraph: prefer "highly susceptible (Table 3)" for consistency and precision (rather than "sensitive").

-p9: the authors state that associations between pfcr K76T and mdAQ and CQ were "expected". Please add citations to support this assertion.

- Table 4: for the patatin-like PO4ase, the median EC50 numbers for the haplotype I1478N / N1480I / N1481S are similar across the columns yet the P value suggests a difference. Please check the figures are all correct. Would IQ ranges have cluttered the Table too much?

- Figure 3: whereas log scales are often used for susceptibility data, I prefer the authors' use of the linear y-axis as in this Figure. Thumbs up from me.

DISCUSSION

- p11: in discussing lumefantrine susceptibility in the context of K13 genotypes, the authors must make clear that other loci not evaluated may be co-selected with K13 and providing this finding i.e. you provide no direct evidence that the K13 variants themselves mediate reduction in lumefantrine susceptibility.

- p15: it is correct to mention a potential threat to management of in-patient paediatric severe malaria, but the authors should also acknowledge that extended artemisinin treatment (6-7 days), but not higher doses, is safe and has prevented recrudescence of K13 mutations in the few studies where this has been evaluated (eg Ashley et al TRAC study; Bethell et al in Cambodia <https://pubmed.ncbi.nlm.nih.gov/21603629/>). This at least theoretically (pending field assessment) supports extended ACT regimens as a possible short-term strategy to manage emerging resistance, such as described here.

- Therefore, can the authors comment on the response needed here to contain parasites with reduced AL susceptibility, while the public health remains low to moderate? Are the ACT plus AQ or Mef ("triple ACT") regimens of Dondorp, ven der Pluijm and colleagues the answer? Or should the sequential ACT approach of Schallig et al (<https://pubmed.ncbi.nlm.nih.gov/29082016/>), compliance challenges notwithstanding given the 6 days of treatment, be trialled in localities where it may be indicated?

signed:

Colin Sutherland

Reviewer #3 (Remarks to the Author):

Artemisinin combination therapies (ACTs) are the current first line treatments for malaria worldwide. The independent emergence and spread of artemisinin resistance in Africa, and the associated loss of efficacy of partner drugs such as lumfantrine (LM) is one of the major challenges controlling and eventually eliminating malaria. Previously confined to South East Asia, the recent discovery of several distinct and relatively common kelch13 mutations which may confer artemisinin resistance in multiple countries of Africa and other continents has sounded the warning bell that artemisinin may lose its effectiveness in Africa, and resistance to partner drugs may follow. Indeed, any loss of efficacy of the partner drugs used in ACTs is alarming especially in Africa which carries ~90% of the worldwide burden of *P. falciparum* malaria.

This manuscript by Tumwebaze et al, presents timely and critical information on two important mutations in the *Plasmodium falciparum* kelch (Pfkkelch) gene, C469Y and A675V that have emerged in Northern Uganda and threatening ACT treatment, and the efficacy of LM in parasite populations in the northern and eastern regions. The study provides important phenotypic characterisation and genotypic data to demonstrate the association of mutations with reduced susceptibility to dihydroartemisinin (DHA) and LM. In addition, this study investigates a comprehensive list of candidate and known drug resistance genes for mutations that could be associated with C469Y and or A675V in conferring reduced LM or DHA sensitivity, which may explain variability in phenotypes of kelch13 mutant parasites. The manuscript is well written and experimentally sound however, a few minor points need to be addressed:

1. Line 34 & 36 –

- The level of ex-vivo drug sensitivity is best assessed relative to the ex-vivo threshold IC50 resistant threshold for each drug rather than between population
- It is not clear to what resistance threshold value the IC50 values are used to conclude the sensitivity level stated here. Therefore, indicate the resistance threshold values for DHA and LM accordingly and then compare between provinces.
- Would the IC50 ex-vivo resistance threshold be 10nM? If so wouldn't the value 2.3nM would be considered ex-vivo susceptible? Please clarify and how this value has been interpreted as being resistant.
- Are the IC50 values in geometric mean or median values. Clearly indicate this in abstract

2. Line 36 – The IC50 resistance threshold for lumefantrine is not well established however values IC50 20-25nM has been mostly considered as the ex-vivo “resistant” threshold. Indicate in text if 20nM is the resistance threshold used in this study.

3. Line 129. No information on these mutations have been provided. Provide information (codon position and amino acid change) on the 15 mutations identified upstream of the propellor domain in the Supplementary info.

4. Line 152 – Include resistance threshold values used for both LM and DHA to reflect the susceptibility levels state here

5. Line 186 - include (PlasmoDB ID) of putative ubiquitin carboxyl-terminal hydrolase 1

6. Line 187 – include PlasmoDB ID of this unknown function gene.

7. Line 214- States that other polymorphisms were strongly associated with either increase or decrease susceptibility to LM and goes on to mention that SNPs in PFMDR1 are associated with increased susceptibility. However, it is not clearly stated in the text for the individual genes putative phospholipase, ApiAP2 transcription factor; MPR1 & MPR2, hemoglobinase, falcipain-2a 3 and plasmepsin 1.

8. Line 254 – Reference the study to which high prevalence of mutations were previously seen.

9. Line 422-427 & 439 – 457 – Lacks references for this methodology

10. Line 437 – What was the concentration range used to test the drugs? Include (maximum to minimum) of the serial dilutions for each for the individual drugs tested, eg. CQ (200 – 10nM), DHA (100 – 10nM). This information is useful for reproducibility

11. Line 514 – Table 2

Multiple mutations in infections have been shown to decrease parasite sensitivity and promote emerging drug resistance. Were these sample/infections carrying one mutation or multiple mutations? It would be useful to include this information as a separate figure such as – a bar chart with the different proportion of mutations per infection (sample) and for each location.

12. Line 525 – Suggest including a column in Table 3 that contains the concentration ranges tested for each drug. Refer to previous comment.

13. Supplementary Tables 2 – 8 lacks p-value significant notation to indicate the significant value.

General formatting

14. Line 39 – space between p and the sign and the equal sign and 6. Similarly for p and equal sign and equal sign and 1.

15. Line 43 – “falciparum” throughout the text should be in italics where appropriate

16. Names of all the genes abbreviated such as Pfk13 or PfCoronin should be in italics

17. Line 52 – Beginning of the sentence so Pfk13 should be written in full “Plasmodium falciparum kelch 13” and not abbreviated

18. Terms “ex/in vivo” throughout the text should be in italics

19. Line 295 typo – prerequisite

20. Line 298-302- something is not right about this sentence

21. In the discussion, could the authors speculate as to the mechanisms by which mutations lead to reduced drug sensitivity

Reviewer 4 - co-reviewer with reviewer 3 (report and comments compiled together)

We thank the reviewers for their thoughtful comments. (Of note, we never received notice of this review by e-mail, but rather saw that the MS was listed as “under revision” on the review website, so proceeded with our revision.) These are addressed point-by-point below. Reviewer summaries that did not include requests for revision have been omitted.

Reviewer #1:

1) Please replace ‘artemisinin resistance’ by ‘artemisinin partial resistance’ that is the official term, in the abstract and the main text.

RESPONSE: We appreciate that WHO prefers the “partial resistance” terminology, as agreed in an international meeting some months ago. We referred to this term in the text, but did not use it throughout, as we feel it is awkward. However, we have acquiesced, and now use “partial resistance” throughout the MS.

2) Last sentence. I suggest adding at the end of the sentence ‘in Northern Uganda’.

RESPONSE: The concluding sentence of the abstract does not mention which region of the country is most at risk, but rather that artemether-lumefantrine is first-line in the country. Further, recent data (some not yet published) shows spread and emergence of key K13 mutations beyond northern Uganda, so we prefer not limiting the geographical scope of this concern, as follows: “Our results raise concern regarding activity of artemether-lumefantrine, the first-line antimalarial in Uganda.”

3) Introduction, Third sentence. I suggest changing the sentence as ‘The resistance phenotype entails ex vivo delayed clearance of parasites after treatment with artemisinins, or in vitro prolonged survival of cultured parasites after exposure to dihydroartemisinin (DHA)’.

RESPONSE: The first half of the sentence concerns a clinical, not lab end-point. For the second half, we think it is clear that “cultured parasites” refers to a lab outcome, which can be ex vivo or in vitro, depending on when cultured parasites are assayed. For clarity and also to address point (1) above, we have modified the sentence to the following. “The partial resistance phenotype entails delayed clearance of parasites after treatment of patients with artemisinins⁴, or prolonged survival of cultured parasites after exposure to dihydroartemisinin (DHA)⁵.”

4) Page 3, second paragraph, first sentence. I suggest to delete ‘amodiaquine’. Amodiaquine resistance in Cambodia is likely linked to the use of chloroquine than artemisinin and I suspect that amodiaquine resistance have emerged long before the introduction of artemisinin.

RESPONSE: Reference to amodiaquine resistance in this context was suggested by SE Asian collaborators in a short review on this subject, but based on this comment we have removed it here, and thus removed reference 26.

5) Page 4. Ref 35. I suggest adding a recent publication (Kubota et al, Front Cell Inf Microb, 2022).

RESPONSE: We have added this reference.

6) Results - Page 7, second paragraph, third sentence. The term ‘delayed clearance’ for in vitro assay is confusing. I suggest to replace by ‘increase survival rate’.

RESPONSE: We agree and have changed “delayed clearance” to “increased survival”.

7) Discussion - Page 11, second paragraph, last sentence. I suggest adding ‘in vitro’ before ‘artemisinin partial resistance’.

RESPONSE: As elsewhere in the MS, we changed “resistance” to “partial resistance”. However, as the in vitro assay is a validated marker for clinical partial resistance, we would prefer not to

add "in vitro" to this sentence; we think that this result shows association with partial resistance, regardless of whether it is measured in vitro or clinically.

8) Page 12, second paragraph. I suggest adding that mutation in Pffalcipain 2a has also been observed in the generation of the F32ART strain (Ariey et al, Nature, 2014). This codon stop occurred sequentially after the Pfk13 476I mutation (S69stop).

RESPONSE: We are happy to expand on this point, especially because our lab has been studying falcipains since the 1990s. We have added a sentence and slightly changed the following sentence, as follows. "Interestingly, a mutation encoding a falcipain-2a stop codon was identified in parasites selected in vitro for resistance to artemisinin, but this selection occurred after emergence of a Pfk13 mutation⁶." Taken together, available data suggest that certain mutations in falcipain-2a can mediate artemisinin resistance independent of or in concert with mutations in Pfk13."

9) Methods - Ex vivo RSA. Could you mention in the manuscript which reference strains were used as controls.

RESPONSE: We did not include laboratory strain controls. Although controls are valuable, we elected not to include them here, due to the extensive work required to maintain and synchronize lab strains for this purpose and because in vitro RSA controls are of limited validity for ex vivo RSAs.

10) Genomic characterization of isolates. Did you use a strain with multiple Pfp2 copy number as control for Pfp2 CNV assessment?

RESPONSE: Yes, we had a control. Thanks for noticing this inadvertent omission. We have added information on this control in our revised sentence in Methods: "The Dd2 strain, which contains amplified *pfdm1* and is single copy for *plasmepsin 2/3*, and the G8 subclone of KH001_053 (kindly provided by Selina Bopp and the TRAC collaboration), which has multiple copies of *plasmepsin 2*⁶⁹, were used as controls.

Reviewer #2:

11) MAJOR COMMENT

The authors target-based approach using MIP genotyping is sound, extensive and provided a wealth of data from a relatively restricted sample set. However only agnostic approaches (genetic crosses OR whole-genome shotgun sequencing and GWAS - requiring larger sample sets) can identify the full complexity of the genetics underpinning both the DHA and lumefantrine susceptibility phenotypes. This needs to be clearly stated.

RESPONSE: We agree fully. To address this point we have added a sixth limitation to the limitations paragraph in the Discussion: "Sixth, our studies were limited to the target genes and not able to characterize the full complement of variation across the genome in relation to drug susceptibility; additional studies, including genome-wide association studies and genetic crosses, will be helpful in this regard."

12) ABSTRACT - second last sentence: the second phrase is difficult to follow. I suggest: "...associated with altered drug response, notably Pfk13 469Y with decreased susceptibility to lumefantrine ($p=6 \times 10^{-8}$) and PfCRT mutations with chloroquine resistance ($p=1 \times 10^{-20}$)."

RESPONSE: We made the requested changes in this sentence

13) - last sentence: unusually, you have understated the impact of your findings here. "raise concern"? Something braver is justified, such as "Our results provide evidence of an emerging threat to artemether-lumefantrine therapeutic efficacy in Uganda."

RESPONSE: We are not usually accused of being too modest! But, we prefer our slightly less “brave” sentence, as there is as yet no clinical evidence that AL is failing in Uganda, and it is reasonably likely that the changes seen to date in Uganda, although worrisome, will not impact on the treatment efficacy of this regimen in the near term.

14) INTRODUCTION - second paragraph: in my view the adj. "massive" is too hyperbolic for a scientific treatise (but not as bad as "huge"). Please substitute with something more precise or appropriate.

RESPONSE: We respectfully disagree. We think that hundreds of million cases and >600,000 deaths per year constitutes a “massive toll”.

15) - third paragraph: yes, 86Y was directly selected in Africa by AQ use but probably more extensive was its indirect selection by CQ, in both cases together with the CVIET haplotype of pfcr. Please try and add this if word count allows.

RESPONSE: The reviewer is discussing selection of the mutation across Africa, but in the relevant sentence we are discussing experimental results; in this context the selection studies were done primarily with AQ, not CQ. We agree that CQ very likely provided the main selective pressure in the field, but there are limited experimental data to prove this, and we think that it will unnecessarily complicate this section of the MS to add mention of the selective pressure of CQ (necessarily with caveats explaining lack of experimental proof).

16) RESULTS - practice in the field is to favour "EC50" over "IC50" currently. If there is a strong rationale for the latter I did not find it (assuming the assays are standard). Can you rule out cytotoxic drug effects as contributing to the apparent inhibition? A brief word in Methods section is justified if so.

RESPONSE: We appreciate the complexities of nomenclature, but EC50 and IC50 are generally used as synonyms, our group has consistently used IC50 in multiple papers, and we hope that we can be consistent in this report. Unfortunately, we have no data to inform regarding potential cytotoxic effects of antimalarials impacting on drug efficacy assays. However, toxicity to host RBCs seems unlikely, as these cells are quite inert.

17) - p5: did the authors find any evidence that a single *P. falciparum* isolate could harbour two or more K13 propeller mutations? Were C469Y and A675V ever seen together?

RESPONSE: This is a good question, but complicated by the fact that most Ugandan isolates are polyclonal. Among all genotyped samples, we identified 4 isolates in which both the 469Y and 675V mutations were detected. In each of these cases, both of the genotypes were mixed (WT plus mutant). It is not possible to determine if any of these samples contained a single clone with both mutations. Considering our inability to determine if any individual strain had >1 mutation, we think it best not to expand on this point in the MS.

18) - p6 bottom paragraph: prefer "highly susceptible (Table 3)" for consistency and precision (rather than "sensitive").

RESPONSE: We made the requested change.

19) p9: the authors state that associations between pfcr K76T and mdaQ and CQ were "expected". Please add citations to support this assertion.

RESPONSE: Due to journal limitations on number of references we have cited a review paper (reference 15) that reviews this subject in detail.

20) Table 4: for the patatin-like PO4ase, the median EC50 numbers for the haplotype I1478N /

N1480I / N1481S are similar across the columns yet the P value suggests a difference. Please check the figures are all correct. Would IQ ranges have cluttered the Table too much?

RESPONSE: We were also surprised by the p-values, but repeated checking has confirmed the statistical analysis. We agree with the reviewer's suggestion that adding IQ ranges to Table 4, though somewhat informative, unacceptably clutters the table, such that the key results are difficult to discern.

21) Figure 3: whereas log scales are often used for susceptibility data, I prefer the authors' use of the linear y-axis as in this Figure. Thumbs up from me.

RESPONSE: Thanks (although this comment actually referred to figure 2).

22) DISCUSSION - p11: in discussing lumefantrine susceptibility in the context of K13 genotypes, the authors must make clear that other loci not evaluated may be co-selected with K13 and providing this finding i.e. you provide no direct evidence that the K13 variants themselves mediate reduction in lumefantrine susceptibility.

RESPONSE: We totally agree, and in fact we doubt that the K13 mutations directly mediate lumefantrine susceptibility. To address this point we added the following sentence to paragraph 7 of the Discussion, where we discuss lumefantrine susceptibility in detail: "However, these results do not indicate a causal role for PfK13 mutations in altered lumefantrine susceptibility."

23) p15: it is correct to mention a potential threat to management of in-patient paediatric severe malaria, but the authors should also acknowledge that extended artemisinin treatment (6-7 days), but not higher doses, is safe and has prevented recrudescence of K13 mutations in the few studies where this has been evaluated (eg Ashley et al TRAC study; Bethell et al in Cambodia <https://pubmed.ncbi.nlm.nih.gov/21603629/>). This at least theoretically (pending field assessment) supports extended ACT regimens as a possible short-term strategy to manage emerging resistance, such as described here.

RESPONSE: We appreciate that extended courses of ACTs may improve treatment outcomes, but these will be for treatment of uncomplicated malaria, not severe malaria (for which speed of action is the key factor). Addressing this point in our Discussion will require a new paragraph, which we think will be a bit off-message. Rather, we think that it will be a high priority to consider longer courses of ACTs if we find that standard courses of ACTs are providing inadequate efficacy. But for now, we respectfully wish to provide a simple message, and so emphasize in our Discussion the primary need for assessments of the efficacies of standard ACTs.

24) Therefore, can the authors comment on the response needed here to contain parasites with reduced AL susceptibility, while the public health remains low to moderate? Are the ACT plus AQ or Mef ("triple ACT") regimens of Dondorp, van der Pluijm and colleagues the answer? Or should the sequential ACT approach of Schallig et al (<https://pubmed.ncbi.nlm.nih.gov/29082016/>), compliance challenges notwithstanding given the 6 days of treatment, be trialled in localities where it may be indicated?

RESPONSE: As above (response 23), we find this consideration of alternative treatment regimens to be very important, but a bit off-message for our MS. We do not think that our data inform regarding the relative merits of triple ACTs, sequential ACTs, longer courses of ACTs, or other strategies. A discussion of this might imply otherwise, and as such would be misleading.

Reviewer #3:

25) Line 34 & 36 –

- The level of ex-vivo drug sensitivity is best assessed relative to the ex-vivo threshold IC50 resistant threshold for each drug rather than between population

- It is not clear to what resistance threshold value the IC50 values are used to conclude the sensitivity level stated here. Therefore, indicate the resistance threshold values for DHA and LM accordingly and then compare between provinces.

- Would the IC50 ex-vivo resistance threshold be 10nM? If so wouldn't the value 2.3nM would be considered ex-vivo susceptible? Please clarify and how this value has been interpreted as being resistant.

- Are the IC50 values in geometric mean or median values. Clearly indicate this in abstract
RESPONSE: Unlike the situation with bacterial drug resistance, there are no standard cut-offs for "resistance" to malaria parasites. Further, for most drugs, including DHA and LM, we see a broad range of susceptibilities, without obvious biphasic (sensitive vs. resistant) phenotypes. Indeed, it is difficult to predict what IC50 level will cause clinical problems; in Cambodia relatively small increases in IC50s for piperazine were accompanied by high failure rates for DHA-piperazine. Overall, we feel that it would not be practical or appropriate to discuss susceptibility results in terms of resistance thresholds. Rather, we think it is most appropriate to present our results and allow readers to draw inferences about potential clinical relevance of the observed data. As noted in tables 3 and 4, IC50s are expressed as medians; we added mention of median measures to the abstract.

26) Line 36 – The IC50 resistance threshold for lumefantrine is not well established however values IC50 20-25nM has been mostly considered as the ex-vivo "resistant" threshold. Indicate in text if 20nM is the resistance threshold used in this study.

RESPONSE: As noted above, there is no established resistance threshold for lumefantrine, and we think that it would be inappropriate to arbitrarily establish a threshold for this MS.

27) Line 129. No information on these mutations have been provided. Provide information (codon position and amino acid change) on the 15 mutations identified upstream of the propeller domain in the Supplementary info.

RESPONSE: In fact, this information was provided. It is available in the last column of Supplemental Table 2 ("Polymorphisms other than K13-propeller domain mutations that are associated with RSA survival"); the mutations listed in the row describing PfK13 are necessarily all non-propeller domain mutations.

28) Line 152 – Include resistance threshold values used for both LM and DHA to reflect the susceptibility levels state here

RESPONSE: As noted above, there are no established resistance thresholds for lumefantrine or DHA, and we think that it would be inappropriate to arbitrarily establish a threshold for this MS.

29) Line 186 - include (PlasmoDB ID) of putative ubiquitin carboxyl-terminal hydrolase 1

RESPONSE: This information is in Supplemental Table 2.

30) Line 187 – include PlasmoDB ID of this unknown function gene.

RESPONSE: This information is in Supplemental Table 2.

31) Line 214- States that other polymorphisms were strongly associated with either increase or decrease susceptibility to LM and goes on to mention that SNPs in PFMDR1 are associated with increased susceptibility. However, it is not clearly stated in the text for the individual genes putative phospholipase, ApiAP2 transcription factor; MPR1 & MPR2, hemoglobinase, falcipain-2a 3 and plasmepsin 1.

RESPONSE: This information is all provided in Table 4, which is cited earlier in this paragraph. For clarity we have cited it again at the end of the relevant sentence.

32) Line 254 – Reference the study to which high prevalence of mutations were previously seen.

RESPONSE: We added a reference providing data from prior years (reference 21).

33) Line 422-427 & 439 – 457 – Lacks references for this methodology

RESPONSE: We added a brief sentence allowing us to reference routine culture methodology described in more detail in a prior study: “Parasites were placed in culture as previously described” (reference 33). The techniques in the following paragraph (beginning on line 431 in our original submission) are already referenced in the first sentence of the paragraph (reference 33), and it is quite clear that this reference refers to the full paragraph.

34) Line 437 – What was the concentration range used to test the drugs? Include (maximum to minimum) of the serial dilutions for each for the individual drugs tested, eg. CQ (200 – 10nM), DHA (100 – 10nM). This information is useful for reproducibility

RESPONSE: Based on this request (and comment 36) we have added a column to Table 3 with this information.

35) Line 514 – Table 2

Multiple mutations in infections have been shown to decrease parasite sensitivity and promote emerging drug resistance. Were these sample/infections carrying one mutation or multiple mutations? It would be useful to include this information as a separate figure such as – a bar chart with the different proportion of mutations per infection (sample) and for each location.

RESPONSE: As noted, most infections in Uganda are polyclonal, making it very difficult to distinguish specific haplotypes. The best that we can do with the available data is to look for associations with individual mutations. Assignment of haplotypes will require more sophisticated analyses, as we now mention in the MS (see response to point 11).

36) Line 525 – Suggest including a column in Table 3 that contains the concentration ranges tested for each drug. Refer to previous comment.

RESPONSE: We added a column to Table 3 with this information, as requested.

37) Supplementary Tables 2 – 8 lacks p-value significant notation to indicate the significant value.

RESPONSE: We are not sure what the reviewer is requesting. P-values are provided in Supplemental Tables 3-8. For these complex multiple-comparison analyses we think that it is not appropriate to assign a cut-off for significance, as the degree of association is more informative than an arbitrary significant/non-significant dichotomy.

General formatting

38) Line 39 – space between p and the sign and the equal sign and 6. Similarly for p and equal sign and equal sign and 1.

RESPONSE: We used space-saving formatting in part to limit our word count in the abstract, as we were interested in best informing readers with valuable data, rather than using up our word count with extra spacing. We checked an older paper from our group in Nat Commun, and the final formatting did not include the spacing requested above. So, appreciating the need to have appropriate final formatting, we have maintained current formatting so that we do not need to reword the abstract to keep within the required word count. Of course, we are agnostic concerning the final formatting, and copy-editors are free to make final adjustments as needed.

39) Line 43 – “falciparum” throughout the text should be in italics where appropriate
RESPONSE: We did italicize correctly. Note that falciparum should not be italicized when used as an adjective (e.g “falciparum malaria”).

40) Names of all the genes abbreviated such as Pfk13 or PfCoronin should be in italics
RESPONSE: We agree. We were careful to italicize names of genes, but NOT names of proteins. We refer to both genes and proteins at different points in the MS, and we have rechecked and taken pains to format both correctly.

41) Line 52 – Beginning of the sentence so Pfk13 should be written in full “Plasmodium falciparum kelch 13” and not abbreviated
RESPONSE: We already did this with first usage of the term (this was apparently missed by our reviewer on line 49 of the original submission).

42) Terms “ex/in vivo” thought the text should be in italics
RESPONSE: We prefer not italicizing these terms, as we do not consider them foreign after use in English for many decades, but based on this request we acquiesce and use italics for these terms.

43) Line 295 typo – prerequisite
RESPONSE: Thank you for catching this. The typo has been corrected.

44) Line 298-302- something is not right about this sentence
RESPONSE: We have carefully read and reread this sentence, and it is not clear to us what is “not right” about it. But, in an attempt to improve clarity we reordered the wording as follows: “In studies of parasites from Senegal, in vitro selection of *P. falciparum* with increasing concentrations of DHA selected for parasites with delayed clearance associated with mutations in Pfcoronin, but not PfK13¹⁰; impacts of Pfcoronin mutations on clearance after DHA exposure varied based on parasite genetic background”.

45) In the discussion, could the authors speculate as to the mechanisms by which mutations lead to reduced drug sensitivity
RESPONSE: Mechanisms by which well-established mutations (in PfCRT, PfMDR1, PfDHFR, PfDHPS, and even PfK13) mediate altered drug susceptibility are quite well understood, and well-covered in a large literature (we cite a few key review papers that summarize this information). Mechanisms by which our newly identified “candidate” mutations might mediate resistance are unknown, but when there are suggestions, these are covered in our Discussion (e.g. “Loss of falcipain activity due to treatment with specific inhibitors or gene knockout markedly blunted the antimalarial activity of DHA, indicating that falcipain-mediated proteolysis of hemoglobin is needed for efficient activation of artemisinins.” “The biological basis of contributions of Pfcoronin mutations to artemisinin resistance is unknown, but it was noted that impacts of Pfcoronin mutations are masked by PfK13 mutations, suggesting that mutations in the two proteins may impact on the same parasite mechanisms”. Additional discussion about other potential mediators, constrained by our limited understanding of their roles, comprises paragraph 6 of the Discussion. Respectfully, we think that additional speculation about mechanisms, based on no data, would not be terribly informative to readers.

46) Unrelated to our review, we discovered during proof-reading that values in the last column of Table 5 were in error. This was corrected and the correction necessitated minor changes to the last paragraph of Results. The changes in text are shown with track changes, but we did not

use track changes for the table to avoid errors in type-setting. Of note, these changes were minor, and did not affect interpretation of results.

REVIEWERS' COMMENTS

Reviewer #2 (Remarks to the Author):

The authors have dealt with all my comments adequately for the most part.

One or two remaining minor comments:

- I support the authors' response to Reviewer 3, that for most malaria drugs there is not an established cut-off between "susceptible" and "resistant"

- Line 61 - "prolonged survival" sounds the same or similar to "delayed clearance" ... and if this refers to RSA estimates of survival then as this is a single time-point assay it would be better to call it "enhanced survival" or an equivalent term.

- Line 63 - "susceptibility" is preferable to "sensitivity"

- Line 130 - to clarify this statement perhaps the following is an improvement: "Neither amplification of pfmdr1 and plasmepsin 2/3 gene copy numbers, nor PfCRT mutations associated with piperazine resistance in Asia (T93S, H97Y, F145I, I218F, M343L, G353V) 40, 41, were seen."

- Figure 3: could the A675V box-plot be the same colour in both panels please?

- Line 256: surely this should read "... and increased parasite survival."

- Line 310: space missing before pfubp1 (could be a pdf conversion error)

Colin Sutherland

Reviewer #3 (Remarks to the Author):

All comments have been addressed satisfactorily

To the Editors:

Thank you for the re-review of our manuscript, entitled "Decreased susceptibility of *Plasmodium falciparum* to both dihydroartemisinin and lumefantrine in northern Uganda." We address the limited comments of the reviewers in a point-by-point manner below.

Reviewer #1: There were no comments from this reviewer.

Reviewer #2

1) The authors have dealt with all my comments adequately for the most part.

RESPONSE: Thank you.

2) I support the authors' response to Reviewer 3, that for most malaria drugs there is not an established cut-off between "susceptible" and "resistant".

RESPONSE: Thank you.

3) Line 61 - "prolonged survival" sounds the same or similar to "delayed clearance" ... and if this refers to RSA estimates of survival then as this is a single time-point assay it would be better to call it "enhanced survival" or an equivalent term.

RESPONSE: We agree. We have changed "prolonged survival" to "enhanced survival" in 6 places in the manuscript.

4) Line 63 - "susceptibility" is preferable to "sensitivity"

RESPONSE: After years working in this area we walk a fine line, using "sensitivity" to discuss the general concept of drug action (as in the sentence on line 63) and "susceptibility" when referring to a specific result for a specific assay. Thus, we have not made a change here. Of note, our text includes 8 usages of "sensitivity" and 49 usages of "susceptibility", and on rereading we believe that the usage has consistently followed the distinction noted above.

5) Line 130 - to clarify this statement perhaps the following is an improvement: "Neither amplification of *pfmdr1* and *plasmepsin 2/3* gene copy numbers, nor PfCRT mutations associated with piperazine resistance in Asia (T93S, H97Y, F145I, I218F, M343L, G353V) 40, 41, were seen."

RESPONSE: Thanks for the suggestion. We have made the suggested change.

6) Figure 3: could the A675V box-plot be the same colour in both panels please?

RESPONSE: This is a good point. We have changed both Figure 2 and Figure 5 so that the same colors are used for the same genotype in panel A and panel B.

7) Line 256: surely this should read " ... and increased parasite survival."

RESPONSE: Thanks very much for identifying this error. Also, for consistency (see response 3) we changed this to "enhanced survival".

8) Line 310: space missing before *pfubp1* (could be a pdf conversion error)

RESPONSE: This error was corrected.

Reviewer #3: All comments have been addressed satisfactorily.

RESPONSE: Thank you.